# Survive and Thrive: Outcomes of Children Enrolled in a Follow-Up Clinic for Small and Sick Newborns in Rural Rwanda

**DOI:** 10.3390/healthcare12232368

**Published:** 2024-11-26

**Authors:** Alphonse Nshimyiryo, Dale A. Barnhart, Mathieu Nemerimana, Kathryn Beck, Kim Wilson, Christine Mutaganzwa, Olivier Bigirumwami, Evelyne Shema, Alphonsine Uwamahoro, Cécile Itangishaka, Silas Havugarurema, Felix Sayinzoga, Erick Baganizi, Hema Magge, Catherine M. Kirk

**Affiliations:** 1Partners In Health/Inshuti Mu Buzima, Kigali P.O. Box 3432, Rwanda; anshimyiryo@pih.org (A.N.); daleaubrey@gmail.com (D.A.B.); mnemerimana@pih.org (M.N.); or kbeck@pih.org (K.B.); or cmutaganzwa@pih.org (C.M.); ebaganizi@pih.org (E.B.); 2Global Health and Social Medicine, Harvard Medical School, Boston, MA 02115, USA; 3Division of General Pediatrics, Boston Children’s Hospital, Boston, MA 02130, USA; kim.wilson@childrens.harvard.edu; 4Rwinkwavu District Hospital, Ministry of Health, Ngoma P.O. Box 48, Rwanda; bigiru200@gmail.com (O.B.); shemeva@gmail.com (E.S.); 5Kirehe District Hospital, Ministry of Health, Kibungo P.O. Box 45, Rwanda; alpros14@gmail.com (A.U.); itacici@gmail.com (C.I.); havugsilas@gmail.com (S.H.); 6Rwanda Biomedical Center, Kigali P.O. Box 7162, Rwanda; felix.sayinzoga@gmail.com; 7Division of Global Health Equity, Brigham and Women’s Hospital, Boston, MA 02115, USA; hema.magge@gmail.com; 8Faculty of Global Health Delivery, University of Global Health Equity, Kigali P.O. Box. 6995, Rwanda

**Keywords:** neurodevelopment, neonatal follow-up, prematurity, early childhood development, developmental monitoring, early intervention, nurturing care, low birthweight, small vulnerable newborns, post-discharge follow-up

## Abstract

Introduction: Children born small or sick are at risk of death and poor development, but many lack access to preventative follow-up services. We assessed the impact of Pediatric Development Clinics (PDC), which provide structured follow-up after discharge from hospital neonatal care units, on children’s survival, nutrition and development in rural Rwanda. Methods: This quasi-experimental study compared a historic control group to children receiving PDC in Kayonza and Kirehe districts. Study populations in both districts included children born preterm or with birthweight < 2000 g and discharged alive. Kirehe additionally included children with hypoxic ischemic encephalopathy (HIE). Home-based cross-sectional surveys were conducted in Kayonza among children with expected chronological age 11–36 months in 2014 (controls) and 2018 (PDC group) and in Kirehe among children with expected chronological age 17–39 months in 2018 (controls) and 2019 (PDC group). Outcomes were measured using anthropometrics and the Ages and Stages Questionnaires. We used weighted logistic regression to control for confounding and differential non-participation. Results: PDC children (*n* = 464/812, 57.1%) were significantly more likely to participate in surveys (83.0% vs. 65.5%), have very low birthweight (27.6% vs. 19.0%), and be younger at the survey (26.2 vs. 31.1 months). 6.9% (*n* = 56) died before the survey. PDC was associated with reduced odds of death (aOR = 0.49, 95% CI: 0.26–0.92) and reduced odds of developmental delay (aOR = 0.48, 95% CI: 0.30–0.77). In Kayonza, PDC was associated with reduced stunting (aOR = 0.52, 95% CI: 0.28–0.98). PDC was not associated with reduced underweight or wasting. Conclusions: PDC was associated with improved survival and development among children born preterm, with low birthweight, or with HIE. Increased access to PDC, scale-up across Rwanda, and implementation of similar services and early intervention in other low-resource settings could support children born small or sick.

## 1. Introduction

One in 10 births globally is premature, resulting in millions of children each year born too soon in addition to millions of others born too small [1,2]. Improvements in coverage and quality of care at birth and specialized care for small and sick newborns have contributed to global progress in reducing neonatal mortality [3]. However, children born preterm or with low birth weight (LBW) are still at increased risk of growth faltering [4], early childhood mortality [5,6], developmental delay and disability [6,7], as well as chronic health conditions [8]. It is estimated that nearly a quarter of preterm or LBW infants and a third of survivors of birth asphyxia in low- and middle-income countries (LMICs) have a neurodevelopmental impairment [9].

To shift the agenda beyond neonatal survival, additional interventions are needed to provide preventative, follow-up care for these high-risk newborns and to allow for early identification of and intervention for any health, nutritional, or developmental concerns. An expert review panel for the World Health Organization and UNICEF identified the need for post-discharge follow-up as one of the essential components for scaling small and sick newborn care in LMICs [10]. In high-income countries, the standard of care is to provide routine follow-up care with a team of specialists [11]. However, in LMICs, there are too few specialists to provide this care and a lack of interventions that cater to the targeted needs of preterm, LBW, and other children at high risk for developmental issues [12]. Despite the growing availability of specialized inpatient neonatal care units (NCUs) in LMICs and evidence of post-discharge follow-up needs [13], there are limited examples of specialized follow-up programs after NCU discharge [14]. There is also limited guidance on the required components of follow-up services [15].

In alignment with global priorities to reduce neonatal mortality, Rwanda has made tremendous progress in reducing neonatal mortality by expanding access to specialized NCUs at hospitals [16]. The neonatal mortality rate declined by 57% from 44 deaths to 20 deaths per 1000 live births for the period from 2000 to 2020 [17]. As has been reported from other LMICs [18], it is expected that the introduction of the use of continuous positive airway pressure (CPAP) machines in neonatology care at hospitals in Rwanda [19,20] has contributed to improved short- and long-term outcomes, including increased survival beyond the neonatal period for preterm and LBW babies. In 2014, the Rwandan Ministry of Health (MOH), Partners In Health/Inshuti Mu Buzima (PIH/IMB), and UNICEF introduced the Pediatric Development Clinic (PDC), a task-shifted model to provide regular medical, nutritional, and developmental follow-up of infants at high risk of poor early childhood outcomes after discharge from the NCU [21]. This study aims to compare the survival, nutritional status, and developmental outcomes of children born preterm/LBW or with hypoxic ischemic encephalopathy (HIE) in pre- and post-PDC implementation periods.

## 2. Materials and Methods

### 2.1. Study Design and Setting

We conducted a quasi-experimental pre–post study to compare outcomes of high-risk children born within the catchment areas of Rwinkwavu District Hospital and Kirehe District Hospital before and after the implementation of the PDC program. Data on the primary outcomes for this study came from four cross-sectional household surveys and a review of medical records in hospital registers and the electronic medical record system for children enrolled in PDC.

Both district hospitals are public hospitals located in rural Kayonza and Kirehe Districts in eastern Rwanda. Rwinkwavu District Hospital serves approximately 200,000 people [22] through a network of 8 outpatient health centers, while Kirehe District Hospital serves approximately 400,000 [22] plus a refugee camp of 60,000 people [23] through a network of 18 outpatient health centers. Prior to the implementation of PDC, specialized NCUs were available at both hospitals with care provided by nurses and general practitioner physicians, and all neonates born preterm, with LBW (<2000 g) and with HIE were referred there for care [24]. NCUs at both district hospitals provide level two neonatal care for small and sick newborns [25], including kangaroo mother care for preterm or LBW babies, nutritional support, management of sepsis, respiratory distress management including CPAP, hyperbilirubinemia, and asphyxia. Since 2005, both hospitals have been supported by PIH/IMB, a non-governmental international organization dedicated to health system strengthening. PIH/IMB’s support included intermittent mentorship to NCU clinical staff by pediatricians.

### 2.2. The PDC Intervention Description

The PDC program provides medical, nutrition, and developmental follow-up to children with high risk of poor early childhood outcomes at hospital- and health center-based PDCs. Children are primarily referred to PDC from hospital NCUs at the time of discharge when they meet the conditions for enrollment in PDC. The PDC is operated by an inter-professional team of nurses and social workers with supervision oversight from a general practitioner, who provide care through outpatient clinics that operate one or two days per week. At the beginning of the PDC assignment, providers receive five days of training on components of PDC care and about once-monthly ongoing post-training mentorship by pediatricians and PDC staff. On clinic days, caregivers and children participate in a group education session on topics such as play and communication, health, nutrition, and hygiene, followed by individual nurse consultations to assess vital signs, danger signs, growth, nutritional intake, and child development. During the period covered by this study, child development was assessed using the Ages and Stages Questionnaires-3 (ASQ-3). PDC provides infant formula when medically necessary for infants under 6 months, food support for mothers of breastfeeding children less than 6 months who are growth faltering, and food support for children over 6 months who are growth faltering. Caregivers may also receive additional social support, counseling, and transport reimbursements based on social worker assessment of socioeconomic need. All PDC visits are recorded into an electronic medical record system. Under the current model, which was implemented in August 2017, children are systematically followed at 1, 2, 4, 6, 9, 12, 18, 24, 30, and 36 months of age, with preterm or LBW infants receiving additional weekly visits until they reach 1 month corrected age. Prior research indicated that the majority (63.5%) of PDC patients respect their PDC follow-up schedules, though at the time, follow up was only required until 24 months of age for children with no other health, nutritional, or developmental concerns [26]. The PDC intervention was initially implemented in Kayonza in April 2014 and later expanded to Kirehe in May 2016. More details about the PDC intervention and its implementation can be reviewed elsewhere [21,27].

### 2.3. Household Surveys

To evaluate the intervention, cross-sectional household surveys were implemented to measure mortality, nutritional, and developmental outcomes before and after the implementation of the PDC intervention in both districts. Participants in the pre-intervention surveys were identified using hospital NCU records and would have been eligible for PDC if the program had existed at the time of their discharge from the hospital NCU. In both Kayonza and Kirehe districts, the surveys targeted children who were discharged alive from the hospital NCU and born preterm or with LBW (i.e., birthweight < 2000 g, as it was the definition of LBW diagnosis for admission in NCU). At the beginning of the PDC implementation in Kayonza, the baseline survey (historical control group) included only preterm or LBW children as this was anticipated to be the primary and largest population served by the new service (PDC). However, by the time of expansion into Kirehe, it was understood that children with birth asphyxia and suspected HIE were a comparably large population served by the clinic, representing about a third of children enrolled in PDC, and were therefore included in Kirehe data collection to assess programmatic impact [21]. In addition, the surveys targeted children who would be aged between 11–36 months in Rwinkwavu and 11–47 months in Kirehe at the time of data collection. The extended age range in Kirehe was included due to anticipated lower participation in the household survey, particularly among the pre-intervention group, as was previously observed in the Rwinkwavu sample [28]. The target age group of survey participants was determined based on the timeline of household surveys for both control and intervention groups in each district as well as the need to allow at least one year of exposure to the intervention for the PDC group. Children from the Mahama refugee camp in Kirehe and from outside the Rwinkwavu and Kirehe District Hospitals’ catchment areas were excluded from all phases of the study. Household surveys for historic controls occurred between November–December 2014 in Rwinkwavu and between October–December 2018 for Kirehe (Figure 1). The post-intervention surveys included children who met the study inclusion criteria and were born in the PDC era. Data collection among PDC infants occurred between May–July 2018 in Rwinkwavu and between May–July 2019 in Kirehe.

Eligible households in all study groups were identified in the community with support from community health workers based on the geographic address, caregiver name, and demographics from health facility records. A team of trained data collectors invited the child’s primary caregiver—defined as the caregiver who spends the most time with the child, which is most often the mother—to participate in an interview. Consent forms were read aloud to eligible caregivers, and a signature was provided for consenting caregivers, or fingerprint for those with limited literacy. Caregivers who provided written consent were interviewed in their homes about household demographics, the child’s health, and the child’s development. Anthropometric assessments were directly measured. Data were collected using KoboCollect software (https://www.kobotoolbox.org/, accessed on 19 November 2024) on tablets.

### 2.4. Study Participants

Children who were selected to participate in the household surveys were eligible for this study. However, due to logistic constraints that affected the time between the implementation of the PDC program and the household surveys, there were substantial imbalances in the ages of children in the pre- and post-PDC intervention groups with children in the post-PDC intervention group being often younger than children in the pre-intervention group. To create more comparable groups, we retrospectively defined additional eligibility criteria based on the child’s condition (diagnosis), date of birth and child’s expected age on the survey date. We used data on the child characteristics at the time of NCU discharge that we extracted from the NCU registers for historic controls or the EMR as they were recorded at the time of enrollment in PDC for the intervention group. In Kayonza, children were eligible for inclusion in our study if they were aged between 11–36 months (age adjusted for days of prematurity) at the time of the household survey and were born between October 2011–October 2013 and April 2015–April 2017 for the historic control and intervention groups, respectively. In Kirehe, we included in our analysis children who were aged between 17–39 months (age adjusted for days of prematurity) at the time of the household survey and were born between May 2015–April 2016 and May 2016–November 2017 the historic control and intervention groups, respectively. In each district, we emphasized that the historic control group only included children born before PDC, while the intervention group only included children born in the PDC era. In both groups, children with any other comorbidities other than prematurity and LBW for Kayonza, and prematurity, LBW and HIE for Kirehe, were excluded from this study. Therefore, not all individuals invited to participate in the household surveys were eligible for the current analysis. In total, 199 (24.5%) individuals who were invited to household surveys were not eligible for inclusion in this analysis.

Children who were included in the original sampling frame of household surveys and who met these inclusion criteria were included in our analysis, even if they were dead or not found for participation in the household surveys. Non-participation in household surveys (NPHS) occurred if the child had relocated outside of the study catchment area or was unable to be located on the available demographic information in the NCU registers or electronic medical record. As described below, inverse probability weighting was used to account for NPHS and competing risk of death.

### 2.5. Outcome Measures

Mortality after NCU discharge was assessed through either confirmed death by community health workers or the primary caregiver during community-follow-up or documentation of death in the electronic medical records (EMR) system for children enrolled in the PDC intervention group. Nutritional status was collected using standard anthropometric measures for height/length and weight and scored based on WHO Growth Standards [29]. We calculated z-scores for height/length-for-age, weight-for-height/length, and weight-for-age and classified children as stunted (low height/length-for-age), wasted (low weight-for-height/length), or underweight (low weight-for-age) if z-scores were two standard deviations below the mean on WHO Growth Standards [30]. Children’s developmental status was measured using the Kinyarwanda language version of the ASQ-3. The ASQ-3 contains 30 age-specific caregiver-report items across five domains of development: communication, gross motor, fine motor, problem solving, and personal social skills. The ASQ-3 was scored categorically based on standard cut-points for the ASQ-3 in each domain as either on-track or potential developmental delay [31]. A child was categorized as potentially delayed overall if he or she had potential delay in at least one domain of the ASQ-3. Because culturally appropriate ASQ-3 cutoffs have not been validated in Rwanda, for our analysis, we also calculated age-specific z-scores for the five ASQ-3 domains (gross motor, fine motor, communication, problem solving, and personal social), which were based on the means and standard deviations reported in a study conducted in somewhat similar settings among typically developing South African and Zambian children [32]. In addition, we also measured the developmental status of children aged 36 months or younger using the short form of the Caregiver Reported Early Development Instrument (CREDI) [33]. The CREDI tool was only included in the household surveys in Kirehe, as it was designed to be culturally neutral and a potential alternative to the ASQ-3 but was not widely available at the time of the study in Kayonza.

### 2.6. Data Analysis

We described and compared child and caregiver characteristics and our primary outcomes of death, nutritional, and developmental status for the intervention and control groups. We described the data using frequencies and percentages for categorical variables and median and interquartile ranges for continuous variables. We defined very LBW (VLBW as birth weight < 1500 g) and pre-term status (<37 weeks’ gestational age or having prematurity as the reason for admission in hospital NCU or referral to PDC), and these variables were extracted from hospital neonatal records for the historic controls and the PDC electronic medical record for the intervention group. We conducted unadjusted bivariate analyses comparing the intervention and pre-intervention periods using Fisher’s exact test for categorical variables and Wilcoxon rank sum tests for continuous variables.

We used weighted linear and logistic regression to assess whether PDC was associated with reduced mortality, improved child nutrition, and improved developmental status while controlling for possible confounders and differential loss to follow-up (LTFU) (i.e., non-participation in household surveys). Differential LTFU was hypothesized to be a major challenge because (a) PDC children had extended contact with the health system and would be easier to locate and (b) data collectors sometimes reported locating the target household but not the child, suggesting that some cases of LTFU were unreported child death. Additionally, since poor nutritional and child development outcomes are associated with increased risk of death, death was hypothesized to act as a competing risk in our analyses of nutritional and child development outcomes. To mitigate these sources of bias, we used inverse probability weighting (IPW), a method that has been used to account for the impact of non-response on outcome estimates in other cohort studies [34]. We calculated weights as the inverse probability of follow-up by fitting a logistic regression that predicted the probability of survey participation. We included all children who were identified as eligible for study participation in both the control and intervention groups during the study period. Our IPW models adjusted for intervention status, district of residence, sex, VLBW, HIE-status, expected age on the follow-up survey date and estimated distance from home to the nearest health facility. For mortality, IPW weights reflected the probability of LTFU, and our final logistic regression model for the association between PDC participation and mortality adjusted for intervention status, district of residence, sex, birthweight, birth condition (born small due to prematurity or LBW, and HIE), and expected age on the follow-up survey date. For all other outcomes, IPW weights reflected the composite probability of death or LTFU and our final models additionally adjusted for the primary caregiver’s education and marital status. Full details on IPW weights can be found in the Appendix A. Our final models for the association between PDC participation and mortality adjusted for intervention status, district of residence, sex, birthweight, birth condition (small or HIE), and expected age on the follow-up survey date. For nutritional and developmental outcomes, we additionally adjusted for the primary caregiver’s education and marital status. Logistic regression was used for the binary outcomes of death, stunting, wasting, underweight, and any potential delay as assessed by the ASQ-3, while linear regression was used for nutritional z-scores and z-scores for the ASQ-3 subdomains. For each outcome, we fit models for the overall population and models stratified by child’s district of residence. As a sensitivity analysis, we have also reported estimates from the unweighted models as well as from a set of models that exclude HIE children from Kirehe district in the Appendix A. We analyzed data using Stata v.15.1 (Stata Corp, College Station, TX, USA).

## 3. Results

In total, 812 children were eligible for this study, including 464 (57.1%) who received the PDC intervention and 348 (42.9%) who were historic controls (Figure 2). Of these 812 children, 56 (6.9%) died before the household survey, 557 (68.6%) participated in the household survey, and 199 (24.5%) were LTFU at the time of the household survey. Children in the PDC intervention group were significantly less likely to be LTFU than children in the historic control group (17.0% vs. 34.5%). Compared to the historic controls, PDC children were significantly more likely to have VLBW (27.6% vs. 19.0%) and to be younger at the time of the household survey (median age 26.2 vs. 31.1 months) but were otherwise similar (Table 1).

In our unadjusted analyses, mortality after NCU discharge was significantly lower among PDC children than among historic controls (6.2% vs. 14.0%, *p* = 0.002) (Table 2). Nutritional status in the PDC children was similar to the historical controls in terms of stunting (59.8% vs. 65.5%, *p* = 0.199), underweight (29.6% vs. 34.9%, *p* = 0.214), and wasting (6.3% vs. 7.9%, *p* = 0.483). However, we observed a statistically significant improvement in the z-scores for height-for-age (median −2.36 vs. −2.56, *p* = 0.037). More children were classified as developmentally on-track on the ASQ-3 in the PDC intervention group, although this difference was not statistically significant (25.3% vs. 18.6%; *p* = 0.072). When comparing z-scores for the subdomains of the ASQ-3, PDC children had significantly higher performance on the fine motor (median z-score −1.49 vs. −2.55; *p* < 0.001), gross motor (median z-score −1.24 vs. −2.21; *p* < 0.001), and personal social (median z-score −0.86 vs. −1.75; *p* < 0.001) domains of the ASQ. Associations with participating in the PDC program and positive outcomes were consistently stronger in Kayonza, where there were also statistically significant improvements in terms of stunting (55.4% vs. 69.3%, *p* = 0.048), having potential developmental delay (61.0% vs. 93.4%, *p* < 0.001), and performance on the communication (median z-score −1.38 vs. −2.24, *p* < 0.001) and problem solving (−0.51 vs. −1.80; *p* < 0.001) domains of the ASQ-3. In Kirehe, mortality was marginally non-significantly different between the PDC group and historic controls (9.9% vs. 16.8%; *p* = 0.073). The rates of stunting, undernutrition and wasting were not significantly different. Children in PDC were more likely to experience any developmental delay on the ASQ-3 (86.4% vs. 73.7%; *p* = 0.007). This increase reflects a strong reduction in problem solving domain (median z-score −1.41 vs. −0.26; *p* < 0.001) but masks significant improvements in the fine motor (median z-score −1.80 vs. −2.51; *p* = 0.024) and personal social (median z-score −1.32 vs. −1.62; *p* = 0.046) domains of the ASQ-3. We also observed significant lower performance among Kirehe PDC children on the CREDI (median score 50.96 vs. 51.54, *p* < 0.001).

After accounting for confounding and LTFU, the PDC intervention was associated with a 51% reduction in the odds of death (OR = 0.49, 95% CI 0.26 to 0.92) and a 52% reduction in the odds of developmental delay (OR = 0.48, 95% CI 0.30 to 0.77) compared to historic controls (Table 3). In the weighted multiple linear regression analysis, PDC was significantly associated with increased average z-scores for the fine motor (β = 0.68, 95% CI 0.45 to 0.92), gross motor (β = 0.51, 95% CI 0.13 to 0.89), and personal social (β = 0.77, 95% CI 0.47 to 1.07) domains of ASQ-3. There was also a marginally significant association between PDC intervention and increased average height-for-age z-scores (β = 0.22, 95% CI −0.001 to 0.44; *p* = 0.051). As in our bivariate analyses, the PDC intervention was associated with greater improvements in Kayonza district, where the intervention was also significantly associated with a 48% reduction in the odds of stunting (OR = 0.52, 95% CI 0.28 to 0.98) and increased average z-scores for height-for-age (β = 0.52, 95% CI 0.19 to 0.86) and for all five sub-domains of the ASQ-3. In Kirehe, the adjusted analysis no longer showed any association between PDC participation and delays on the ASQ-3, however, the z-scores for the communication and problem-solving domains of the ASQ-3 were still significantly worse in the intervention group.

From the sensitivity analysis, with the exclusion of 150 children with HIE from the Kirehe sample, there was a significantly lower mortality among children who received the PDC intervention in both Kirehe and Kayonza districts in the unadjusted analysis (see Appendix A). The PDC intervention also became significantly associated with improved performance on all domains of the ASQ-3 in the adjusted analysis (Appendix A), however other findings were not substantively altered.

## 4. Discussion

This study assessed the impact of outpatient PDCs on survival, nutritional, and developmental outcomes among children born preterm, LBW, or with HIE in Kayonza and Kirehe districts in rural Rwanda. Our study found that the PDC significantly reduced mortality and improved children’s development but did not significantly improve children’s nutrition, and we found differential impact in the two study districts.

While mortality occurred in both the intervention and historical control groups, our findings indicate that the PDC intervention halved the odds of death after adjusting for confounding and differential survey participation. The PDC intervention package includes the provision of regular medical follow-up to identify and treat all causes of mortality—similar interventions among at-risk children have shown to be effective in reducing the risk of death [35].

Similarly, we observed a reduction in the odds of potential developmental delay in the intervention group and an association of PDC with a significant increase in the z-scores of the fine motor, gross motor and personal social domains of the ASQ-3. The PDC intervention provides caregivers with general counseling on early childhood development and emphasizes the importance of responsive care and early learning through play and communication for young children through counseling provided in child-friendly spaces at the health facilities, which are proven interventions to support children’s development [36].

That said, while the PDC group reported better developmental outcomes than the control group, a high prevalence of potential developmental delays persisted among children exposed to PDC. This finding is consistent with evidence that children born preterm, LBW or with HIE in LMICs are at high risk for developmental issues [9]. A complementary explanation for this high prevalence is that the ASQ-3 is a screening tool, not a diagnostic tool. Furthermore, the ASQ-3 uses binary cutoffs that have not been validated in Rwanda and may not be as sensitive to subtle improvements in children’s developmental status as a culturally adapted continuous metric [37]. Interestingly, when we assessed child development using continuous ASQ-3 z-scores, we did observe significant or marginally significant improvements in the overall study sample across all five developmental domains. In practice, PDC staff felt that the ASQ-3 provided insufficient guidance for individualized counseling. In 2019, the PDC program replaced the ASQ-3 with the International Guide for Monitoring Child Development, a tool which provides monitoring and early intervention for developmental difficulties and is ideal for use in a primary health care setting and has been validated in diverse settings [38,39]. In addition, a more robust model of early intervention using the BabyUbuntu program was adopted given the high number of children who would benefit from additional support. However, this occurred after the study period.

In the overall study population, we did not observe significant differences in the prevalence of stunting, underweight, or wasting in the intervention and control groups. Although we did not observe any significant association between PDC and changes in weight-for-age and weight-for-height z-scores, we did observe a significant improvement in the unadjusted analysis for the height-for-age z-scores and a marginally significant improvement in the fully adjusted models. These marginal improvements in the height-for-age z-scores, but non-significant changes in the prevalence for stunting, may reflect improvements in height-for-age that were not sufficient to shift children into the normal height-for-age category, which would occur—for example—if the intervention shifted a child from the severely stunted to moderately stunted category. Additionally, there was likely a reduction in power from using a binary rather than a continuous outcome. Unlike stunting, which reflects long-term nutritional status, wasting is more vulnerable to seasonal change and shocks and may not fully reflect the impacts of a long-term growth monitoring intervention [40,41]. Furthermore, the risk of wasting was low at baseline, and it may have been difficult to further reduce this proportion in a setting where food insecurity is common [42]. Underweight, which can be viewed as a composite outcome of stunting and wasting, could have been difficult to change for similar reasons, especially given the marginal changes in height-for-age [43]. Challenges implementing nutritional interventions were observed during the PDC program, which may have contributed to these results. In 2017, the nutrition protocol for PDC was modified to improve assessment and management of feeding difficulties in infants utilizing the management of small and nutritionally at-risk infants under six months and their mothers (MAMI) tool [44]. In 2019, a specialized feeding assessment and interventions for children with feeding difficulties were added to address persistent concerns about poor feeding and growth [45]. An mHealth tool was also implemented to address the documented challenges in the quality of growth monitoring in PDC [27] and was associated with improvements in both stunting and underweight [27]. Most of these improvements would not have been fully implemented at the time of the study.

In Kayonza, we observed larger effect sizes, including a significant improvement in height-for-age z-scores, a corresponding reduction in the odds of stunting, and significant improvements in z-scores for all five of the ASQ-3 domains. These results may reflect a combination of the PDC program implementation and study design-related factors. At the time of the household surveys for the PDC intervention group in Kayonza, the PDC program had been established in the district for over a year, and decentralization to health centers had already occurred. In contrast, household surveys in Kirehe occurred in the very first year of PDC services in that district and prior to PDC decentralization at health centers. Consequently, study participants in Kirehe likely experienced more congested clinics and were served by providers who had received less mentorship, likely leading to lower quality of service delivery and a lower likelihood of completing routine clinic visits. Finally—although the age, birthweight, and diagnoses among children in Kayonza were very similar pre- and post-intervention—in Kirehe, children in the intervention group were significantly more likely to be younger and have a very low birth weight (<1500 g). Although we adjusted for these confounders in our models, any residual confounding by these factors would bias our results against the intervention in Kirehe. Additionally, other socioeconomic factors might have facilitated the effectiveness of PDC intervention in favor of Kayonza. For example, findings from national demographic and health surveys conducted around our study period indicate that households in Kayonza were more likely to have a place for washing hands (33.5% vs. 2.7% in Kirehe), while women age 15–49 years were more likely to attend secondary school or higher levels of education (19.5% vs. 12.9% in Kirehe) [17,46]. Our results are consistent with the reported respective decreasing and increasing trends in the prevalence of stunting in Kayonza (down from 42.4% to 28.3%) and Kirehe (increase from 29.4% to 31.3%) among the general population of children under 5 between the years 2014/15 and 2019/20 [17,46].

This study has a number of limitations. First, the use of a quasi-experimental design, comparing outcomes before and after the PDC intervention is appropriate, as the random assignment of children to the intervention would be unethical. The design is also suitable for assessing the real-world effectiveness of healthcare interventions. However, the absence of randomization could have introduced biases due to unmeasured differences between the control and intervention groups. Unequal distribution of unmeasured confounding variables of the association between the intervention and outcomes would have resulted in either underestimation or overestimation of the effect size of PDC. Second, as with all pre-post studies, our study is vulnerable to concurrent interventions and secular trends; however, the staggered implementation across districts mitigates this risk and we are not aware of comparable interventions targeting child development in these districts, although there were concurrent interventions to improve the quality of NCUs at hospitals which are supported by PIH/IMB [47]. Third, the PDC intervention was actively evolving during and after the study period, and the quality of the intervention likely improved over time. Improvements in the PDC program have included the strengthened mentoring of PDC nurses and social workers with the support of pediatricians trained in the United States of America, the addition of routine PDC visits at 30 and 36 months, improved child development assessments and interventions, and improved nutrition interventions. Consequently, we would expect the current analysis to underestimate the impact of the current PDC intervention package. Fourth, we retrospectively created additional inclusion criteria to create more comparable intervention and historic control groups; however, ideally these criteria would have been implemented prospectively. Even after implementing these criteria, the intervention group was significantly younger and had a lower birthweight and residual confounding by these factors would bias our results against the intervention. Fifth, this analysis had a sample size of 558 for nutritional and developmental outcomes and may not have been significantly powered for clinically significant improvements. Further, the lack of locally validated developmental assessments is also a limitation. The ASQ-3 was selected based on prior use in Rwanda and availability of regional data for z-score analysis; the addition of the CREDI in Kirehe was also based on best available global tools at the time of this research. Finally, the use of IPW method to account for the impact of LTFU on outcome estimates could have yielded robust results, though it may still not fully eliminate the bias introduced by LTFU. Children were significantly more likely to be LTFU in the control group (35%) than in the intervention group (17%)—this differential LTFU would have led to an underestimation of the effect size of PDC if many cases of LTFU were unreported child deaths or children with poor nutritional and/or developmental outcomes.

## 5. Conclusions

Our findings suggest that an outpatient intervention that provided regular growth monitoring, nutritional education, monitoring of development, counseling on responsive care and early learning, and monitoring and treatment of illnesses was effective in improving the survival and developmental status of children born preterm, LBW, or with HIE. As evidenced by differences in outcomes between the two districts, sustained mentoring and continued investment in these programs may be necessary to ensure high-quality programs capable of generating consistent improvements across multiple domains of child development. Implementation and evaluation at-scale of PDC services in Rwanda and the integration of similar services into health systems in other low-resource settings may be feasible and impactful to save and improve lives of children born small and/or sick. Additionally, we recommend that future scaling-up of the PDC intervention should include a plan for more robust evaluation that will effectively measure the impact of the improved PDC, employing more appropriate research designs, such as a cluster randomized controlled trial and tools that help to address limitations highlighted in this study.

## Figures and Tables

**Figure 1 healthcare-12-02368-f001:**
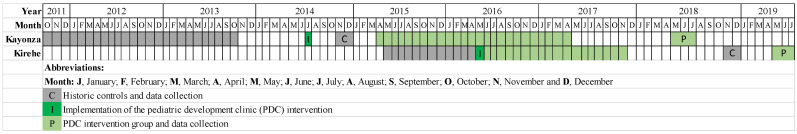
Study period and timeline of household surveys.

**Figure 2 healthcare-12-02368-f002:**
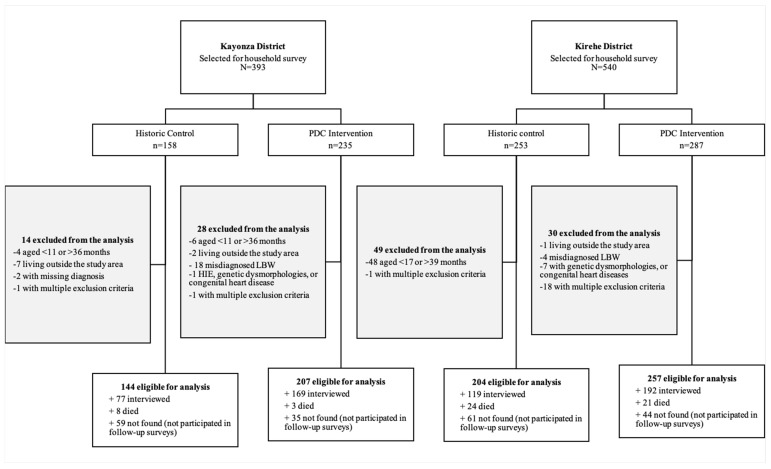
Study participant flow diagram. PDC, pediatric development clinic; LBW (low birth weight) condition, but birthweight > 2050 g; HIE, hypoxic ischemic encephalopathy.

**Table 1 healthcare-12-02368-t001:** Characteristics of children eligible for the study (*n* = 812).

Variable	Kayonza District, *n* = 351			Kirehe District, *n* = 461			All, *n* = 812		
	Historical Control 144 (41.0%)	PDC Intervention 207 (59.0%)	*p*-Value	Historical Control 204 (44.2%)	PDC ^1^ Intervention 257 (55.8%)	*p*-Value	Historical Control 348 (42.9%)	PDC Intervention 464 (57.1%)	*p*-Value
Child sex, *n* (%)			0.938			0.345			0.406
Male	66 (45.8)	94 (45.4)		113 (55.4)	131 (51.0)		179 (51.4)	225 (48.5)	
Female	78 (54.2)	113 (54.6)		91 (44.6)	126 (49.0)		169 (48.6)	239 (51.5)	
Birth weight (grams), *n* (%)			0.478			0.001			0.003
≥1500	104 (72.2)	141 (68.1)		178 (87.2)	192 (74.7)		282 (81.0)	333 (71.8)	
<1500	40 (27.8)	66 (31.9)		26 (12.8)	62 (24.1)		66 (19.0)	128 (27.6)	
Missing data	0 (0.0)	0 (0.0)		0 (0.0)	3 (1.2)		0 (0.0)	3 (0.7)	
Diagnosis, *n* (%)			NA			0.104			0.082
Preterm/LBW ^2^	144 (100.0)	207 (100.0)		145 (71.1)	190 (73.9)		289 (83.0)	397 (85.6)	
HIE ^3^	NA	NA		57 (27.9)	58 (22.6)		57 (16.4)	58 (12.5)	
HIE ^3^ & Preterm/LBW ^2^	NA	NA		2 (1.0)	9 (3.5)		2 (0.6)	9 (1.9)	
Child’s expected age on survey date (months), median [IQR ^4^]	22.4 [17.8–28.2]	25.7 [18.9–29.5]	0.162	34.6 [31.1–37.0]	26.7 [23.6–32.0]	<0.001	31.1 [25.7–35.4]	26.2 [22.0–30.4]	<0.001
Primary caregiver’s education ^5^, *n* (%)			0.262			0.684			0.620
None	19 (24.4)	28 (16.8)		19 (16.1)	37 (19.3)		38 (19.4)	65 (18.1)	
Primary	52 (66.7)	115 (68.9)		82 (69.5)	124 (64.6)		134 (68.4)	239 (66.6)	
Secondary/higher	7 (9.0)	24 (14.4)		17 (14.4)	31 (16.1)		24 (12.2)	55 (15.3)	
Primary caregiver’s marital status ^5^, *n* (%)			0.301			0.290			0.937
Married	34 (43.6)	56 (33.5)		53 (44.9)	101 (52.6)		87 (44.4)	157 (43.7)	
Cohabitating	29 (37.2)	77 (46.1)		47 (39.8)	60 (31.2)		76 (38.8)	137 (38.2)	
Single/Widowed/Divorced	15 (19.2)	34 (20.4)		18 (15.3)	31 (16.2)		33 (16.8)	65 (18.1)	

^1^ PDC, pediatric development clinic; ^2^ LBW, low birth weight (<2000 g); ^3^ HIE, hypoxic ischemic encephalopathy; ^4^ IQR, interquartile range and ^5^ data only collected from among children who participated in the home-based follow-up surveys.

**Table 2 healthcare-12-02368-t002:** Outcomes among observed children.

Variable	Kayonza District, *n* = 351			Kirehe District, *n* = 461			All, *n* = 812		
	Historical Control	PDC Intervention	*p*-Value	Historical Control	PDC Intervention	*p*-Value	Historical Control	PDC Intervention	*p*-Value
Death, *n* = 613, *n* (%)			0.007			0.073			0.002
No	77 (90.6)	169 (98.3)		119 (83.2)	192 (90.1)		196 (86.0)	361 (93.8)	
Yes	8 (9.4)	3 (1.7)		24 (16.8)	21 (9.9)		32 (14.0)	24 (6.2)	
Stunting, *n* = 547 ^1^, *n* (%)			0.048			0.903			0.199
No	23 (30.7)	75 (44.6)		44 (37.0)	67 (36.2)		67 (34.5)	142 (40.2)	
Yes	52 (69.3)	93 (55.4)		75 (63.0)	118 (63.8)		127 (65.5)	211 (59.8)	
Underweight, *n* = 556 ^1^, *n* (%)			0.178			0.708			0.214
No	48 (63.2)	122 (72.2)		79 (66.4)	132 (68.8)		127 (65.1)	254 (70.4)	
Yes	28 (36.8)	47 (27.8)		40 (33.6)	60 (31.2)		68 (34.9)	107 (29.6)	
Wasting, *n* = 540 ^1^, *n* (%)			0.621			0.610			0.483
No	65 (90.3)	151 (92.1)		111 (93.3)	176 (95.1)		176 (92.1)	327 (93.7)	
Yes	7 (9.7)	13 (7.9)		8 (6.7)	9 (4.9)		15 (7.9)	22 (6.3)	
ASQ-3 total, *n* = 549 ^1^, *n* (%)			<0.001			0.007			0.072
Typical	5 (6.6)	64 (39.0)		31 (26.3)	26 (13.6)		36 (18.6)	90 (25.3)	
Potential delay	71 (93.4)	100 (61.0)		87 (73.7)	165 (86.4)		158 (81.4)	265 (74.7)	
**Anthropometric z-scores ^2^**									
Height-for-age, *n* = 547, median [IQR]	−2.61 [−3.44, −1.79]	−2.19 [−2.99, −1.29]	0.005	−2.52 [−3.09, −1.70]	−2.44 [−3.09, −1.53]	0.816	−2.56 [−3.23, −1.77]	−2.36 [−3.04, −1.37]	0.037
Weight-for-age, *n* = 556, median [IQR]	−1.40 [−2.37, −0.54]	−1.31 [−2.15, −0.57]	0.430	−1.66 [−2.25, −0.80]	−1.48 [−2.31, −0.68]	0.646	−1.54 [−2.33, −0.66]	−1.37 [−2.22, −0.58]	0.312
Weight-for-height, *n* = 541, median [IQR]	0.01 [−0.93, 0.61]	−0.18 [−1.11, 0.39]	0.319	−0.15 [−0.93, 0.50]	−0.28 [−0.91, 0.50]	0.915	−0.12 [−0.93, 0.57]	−0.22 [−1.00, 0.43]	0.576
**ASQ-3 z-scores by domain ^3^**									
Fine Motor, *n* = 536, median [IQR]	−2.68 [−3.10, −1.79]	−1.33 [−2.06, −0.34]	<0.001	−2.51 [−3.47, −1.31]	−1.80 [−2.91, −1.04]	0.024	−2.55 [−3.24, −1.51]	−1.49 [−2.70, −0.75]	<0.001
Gross Motor, *n* = 554, median [IQR]	−2.09 [−3.65, −1.06]	−1.15 [−2.03, 0.03]	<0.001	−2.23 [−3.63, −0.62]	−1.35 [−3.45, −0.35]	0.092	−2.21 [−3.63, −0.77]	−1.24 [−2.58, −0.09]	<0.001
Communication, *n* = 554, median [IQR]	−2.24 [−3.24, −1.59]	−1.38 [−2.61, −0.13]	<0.001	−1.96 [−3.79, −0.66]	−2.52 [−4.06, −0.98]	0.297	−2.10 [−3.34, −0.98]	−2.03 [−3.14, −0.54]	0.056
Problem solving, *n* = 551, median [IQR]	−1.80 [−3.07, −0.64]	−0.51 [−1.86, 0.25]	<0.001	−0.26 [−1.64, 0.13]	−1.41 [−2.75, −0.51]	<0.001	−0.93 [−2.27, 0.13]	−0.96 [−2.32, −0.13]	0.296
Personal social, *n* = 554, median [IQR]	−1.81 [−3.17, −0.77]	−0.65 [−1.48, 0.34]	<0.001	−1.62 [−2.70, −0.73]	−1.32 [−2.71, −0.37]	0.046	−1.75 [−3.11, −0.73]	−0.86 [−2.04, 0.02]	<0.001
CREDI ^4^ Overall score, median [IQR], *n* = 262	-	-	-	51.54 [51.15, 51.95]	50.96 [50.36, 51.51]	<0.001	51.54 [51.15, 51.95]	50.96 [50.36, 51.51]	<0.001

PDC, pediatric development clinic; IQR, interquartile range. ^1^ In 557 children who participated in the homebased survey were eligible for analysis. Ten children were missing data on stunting for either biologically implausible height-for-age or missing data on height. For underweight, 1 child had biologically implausible weight-for-age data. For wasting, 17 children had missing data, including 12 children with biologically implausible weight-for-height and 5 with missing height. In total, 8 children had missing data for the Ages and Stages Questionnaire (ASQ) total score (1 whose true age differed from the administered ASQ survey by more than 3 months and 7 children who completed fewer than 4 out of 6 questions on at least one ASQ sub-domain). ^2^ Z-scores were calculated using the age-specific means and standard deviations from the WHO’s Child Growth Standards. ^3^ Z-scores were calculated based on the means and standard deviations reported in study of typically developing South African and Zambian children [32]. ^4^ The Caregiver Reported Early Development Index (CREDI) was used only in Kirehe, as it was designed to be a more culturally neutral assessment of child development. However, it was not widely available at the time of the study in Kayonza. Fisher’s exact test was used for categorical variables, while the Wilcoxon rank sum test was used for continuous variables.

**Table 3 healthcare-12-02368-t003:** Association between enrollment in the pediatric development clinic program and outcomes, accounting for differential death and non-participation in follow-up surveys.

	Kayonza District		Kirehe District		All	
Binary Outcomes	aOR (95% CI)	*p*-value	aOR (95% CI)	*p*-value	aOR (95% CI)	*p*-value
Death ^1^ (*n* = 613)	0.12 (0.03, 0.50)	0.003	0.62 (0.25, 1.55)	0.309	0.49 (0.26, 0.92)	0.025
Stunting ^2^ (*n* = 544)	0.52 (0.28, 0.98)	0.044	1.17 (0.62, 2.19)	0.624	0.87 (0.59, 1.28)	0.465
Underweight ^2^ (*n* = 553)	0.76 (0.41, 1.39)	0.370	1.19 (0.61, 2.34)	0.605	0.77 (0.52, 1.15)	0.209
Wasting ^2^ (*n* = 537)	1.25 (0.41, 3.84)	0.699	0.35 (0.07, 1.78)	0.207	0.72 (0.35, 1.48)	0.377
ASQ-3 Total ^2^ (*n* = 546)	0.10 (0.04, 0.29)	<0.001	1.31 (0.60, 2.85)	0.497	0.48 (0.30, 0.77)	0.002
**Continuous Outcomes**	**β (95% CI)**	***p*-value**	**β (95% CI)**	***p*-value**	**β (95% CI)**	***p*-value**
**Nutrition indicators**						
Height-for-age z-score (*n* = 544)	0.52 (0.19, 0.86)	0.002	0.04 (−0.35, 0.42)	0.856	0.22 (−0.001, 0.44)	0.051
Weight-for-age z-score (*n* = 553)	0.12 (−0.24, 0.49)	0.511	−0.03 (−0.41, 0.35)	0.885	0.12 (−0.11, 0.34)	0.306
Weight-for-height z-score (*n* = 537)	−0.27 (−0.61, 0.07)	0.123	0.17 (−0.20, 0.54)	0.377	0.05 (−0.17, 0.27)	0.664
**Development indicators**						
ASQ-3 Fine Motor z-score (*n* = 533)	1.09 (0.77, 1.42)	<0.001	−0.16 (−0.55, 0.23)	0.416	0.68 (0.45, 0.92)	<0.001
ASQ-3 Gross Motor z-score (*n* = 551)	1.25 (0.74, 1.75)	<0.001	−0.39 (−1.18, 0.39)	0.326	0.51 (0.13, 0.89)	0.008
ASQ-3 Communication z-score (*n* = 551)	0.80 (0.42, 1.18)	<0.001	−0.69 (−1.33, −0.04)	0.037	0.31 (−0.02, 0.63)	0.064
ASQ-3 Problem Solving z-score (*n* = 548)	0.78 (0.44, 1.11)	<0.001	−0.83 (−1.20, −0.45)	<0.001	0.21 (−0.03, 0.45)	0.081
ASQ-3 Personal Social z-score (*n* = 551)	1.10 (0.64, 1.57)	<0.001	−0.14 (−0.63, 0.34)	0.557	0.77 (0.47, 1.07)	<0.001
CREDI Overall score (*n* = 261)	-	-	−0.04 (−0.28, 0.21)	0.778	−0.04 (−0.28, 0.21)	0.778

^1^ Model was adjusted for lost to follow-up using inverse probability calculated based on intervention status, district of residence, sex, birthweight, condition (small baby/HIE), expected age at the follow-up survey date and estimated distance from home village to the nearest health facility. A weighted logistic regression model was adjusted for the same covariates. ^2^ Model was adjusted for using inverse probability weights calculated based on intervention status, district of residence, sex, birthweight, condition (small baby/HIE), expected age at the follow-up survey date and estimated distance from home village to the nearest health facility. A weighted logistic regression model (for binary outcomes) or linear regression model (for continuous outcomes) was adjusted for intervention status, district of residence, sex, birthweight, condition (small baby/HIE), expected age at the follow-up survey date, primary caregiver’s education level, and marital status.

## Data Availability

The data presented in this study are available on request from the corresponding author and in accordance with Rwanda’s law No. 058/2021 of 13/10/2021 on the protection of personal data and privacy in Rwanda.

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
