# Peer review of "Survive and Thrive: Outcomes of Children Enrolled in a Follow-Up Clinic for Small and Sick Newborns in Rural Rwanda"

_healthcare, 2024, doi:10.3390/healthcare12232368_

Round 1
Reviewer 1 Report
Comments and Suggestions for Authors
Good article with a lot of important detail. Fills a much needed gap and highlights the need for high quality follow-up and management of premature/LBW and neonates suffering from asphyxia A little bit wordy making parts of it hardier to read so could potentially be shortened but not essential.
Author Response
Dear reviewer,
Thank you for taking the time to review our manuscript.
Thank you for the opportunity to revise our manuscript. We have responded to each of the comments below.
We feel the helpful feedback from yourself and other reviewers has improved the manuscript.
Sincerely,
Catherine Kirk on behalf of the authors
Response to Reviewer Comments
Comment 1: Good article with a lot of important detail. Fills a much needed gap and highlights the need for high quality follow-up and management of premature/LBW and neonates suffering from asphyxia.
Response 1: Thank you so much for your feedback!
Comment 2: A little bit wordy making parts of it hardier to read, so could potentially be shortened but not essential.
Response 2: Thank you very much for your feedback. We have now made some necessary edits to make the paper easier to read.
Reviewer 2 Report
Comments and Suggestions for Authors
File attached.

Author Response
Reviewer 2:
Dear reviewer,
Thank you for taking the time to review our manuscript.
Thank you for the opportunity to revise our manuscript. We have responded to each of the comments below.
We feel the helpful feedback from yourself and other reviewers has improved the manuscript.
Sincerely,
Catherine Kirk on behalf of the authors
Response to Reviewer Comments
Comment 1: In the introduction provide a more explicit link between global challenges and Rwanda's approach to addressing them to improve overall flow.
Response 1: Thank you for this comment. We added some additional context to the background, including a World Health Organization-UNICEF expert statement that identifies post-discharge follow-up as one of ten essential components for scaling small and sick newborn care in LMICs (see lines 55-58), as well as a very recent paper on post-discharge follow-up needs identified in Kenya (see lines 62-64).
Comment 2: neonatal mortality reduction in Rwanda (57% decline from 44 to 20 deaths per 1,000 live births) is good, further elaboration on the factors contributing to this reduction, particularly the impact of CPAP machines and the Pediatric Development Clinic (PDC), is needed strengthen the evidence.
Response 2: Thank you very much for pointing this out. We have now re-arranged the text around the reduction in neonatal mortality and possible explanations in Rwanda. We have also cited papers that reported on CPAP implementation in Rwandan hospitals and clarified that the PDC intervention is only provided as a follow-up care after discharge from the NCU.
Comment 3: The use of standardized tools (ASQ-3 and WHO Growth Standards) ensures the consistency and comparability of these measures, though the lack of local validation of ASQ-3 cut-offs for Rwanda is a limitation.
Response 3: Thank you. We recognize this is an important and common challenge in child development research. We acknowledged in our methods the challenge of measuring child development using the ASQ-3 tool which was not yet validated in Rwanda – we provided in the methods what we did as sensitivity analysis to overcome this challenge. Our sensitivity analysis for developmental outcomes was summarized in the text as follows: “Because culturally appropriate ASQ-3 cutoffs have not been validated in Rwanda, for our analysis we also calculated age-specific z-scores for the five ASQ-3 domains (gross motor, fine motor, communication, problem solving, and personal social) which were based on the means and standard deviations reported in a study conducted in kind of similar settings among typically developing South African and Zambian children (see lines 219-224).” Further, the lack of locally validated developmental assessments is also a limitation. The ASQ was selected based on prior use in Rwanda and availability of regional data for z-score analysis; the addition of the CREDI in Kirehe was also based on best available global tools at the time of this research.”
Comment 4: The use of inverse probability weighting (IPW) to account for differential loss to follow-up and death strengthens the analysis, though it may still not fully eliminate the bias introduced by LTFU.
Response 4: Thank you very much for pointing this out. We have now included differential loss to follow-up between intervention and control groups as a limitation for our study (see lines 170-176). It reads as follows: “Finally, the use of IPW method to account for the impact of LTFU on outcome estimates could have yielded robust results, though it may still not fully eliminate the bias introduced by LTFU. Children were significantly more likely to be LTFU in the control group (35%) than in the intervention group (17%) – this differential LTFU would have led to an underestimation of the effect size of PDC if many cases of LTFU were unreported child deaths or children with poor nutritional and/or developmental outcomes.”
Comment 5: The retrospective adjustments to eligibility criteria, particularly for age and comorbidities, ensure better comparability between groups. However, this can also introduce potential biases or limit the generalizability of the results due to the exclusion of certain children.
Response 5: Thank you for bringing this up. Yes, we reported that the retrospective adjustments to study inclusion criteria rather than doing it prospectively was a limitation (see lines 160-164). The text reads as follows: “Fourth, we retrospectively created additional inclusion criteria to create more comparable intervention and historic control groups; however, ideally these criteria would have been implemented prospectively. Even after implementing these criteria, the intervention group was significantly younger and had a lower birthweight and residual confounding by these factors would bias our results against the intervention.”
Comment 6: Adherence to this schedule by caregivers (63.5%) suggests a potential challenge in retaining participants, which could introduce selection bias.
Response 6: Thank you very much for pointing this out. We have now clarified in the methods where we define study participants that the recruitment of study participants for both the control and PDC groups was based on the child data at the time of NCU discharge for infants born during the study period (see lines 178-181). Therefore, adherence to the PDC schedules did not affect the chance of study participation for the intervention group. However, we do recognize that the retention study that showed 63.5% of caregivers respected the visit schedule is an opportunity for programmatic improvement for the PDC that could contribute to improved outcomes for children in the PDC in the future. As we have noted in the discussion, many programmatic improvements have been made and future research can help to understand the impact of the improved program based on a decade in lessons learned from implementation.
Comment 7: The use of a quasi-experimental design, comparing outcomes before and after the PDC intervention, is suitable for assessing the real-world impact of healthcare interventions. However, the absence of randomization can introduce biases related to baseline differences between the pre- and post-intervention groups.
Response 7: Thank you for pointing this out. We have now included our study design as a limitation itself (see lines 141-148). The text reads as follows: “First, the use of a quasi-experimental design, comparing outcomes before and after the PDC intervention is appropriate, as the random assignment of children to the intervention would be unethical. The design is also suitable for assessing the real-world effectiveness of healthcare interventions. However, the absence of randomization could have introduced biases due to unmeasured differences between the control and intervention groups. Unequal distribution of unmeasured confounding variables of the association between the intervention and outcomes would have resulted in either underestimation or overestimation of the effect size of PDC.”
Comment 8: Expand the discussion of the differential impact between the two districts. find possible causes beyond decentralization and mentorship, such as socioeconomic, infrastructural, or community engagement differences. This would provide a more comprehensive interpretation of why the intervention may have been more effective in one district.
Response 8: Thank you for this feedback. We have now added more discussion of the possible explanations for the observed difference between Kayonza and Kirehe (in terms of effectiveness of PDC) (see lines 131-140). The added interpretation reads as follows: “Additionally, other socioeconomic factors might have facilitated the effectiveness of PDC intervention in favor of Kayonza. For example, findings from national demographic and health survey conducted around our study period indicate that households in Kayonza were more likely to have a place for washing hands (33.5% vs. 2.7% in Kirehe), while women age 15-49 years were more likely to attend secondary school or higher level of education (19.5% vs. 12.9% in Kirehe).46 Our results are consistent with the reported respective decreasing and increasing trends in the prevalence of stunting in Kayonza (down from 42.4% to 28.3%) and Kirehe (increase from 29.4% to 31.3%) among the general population of children under 5, between the years 2014/15 and 2019/20.14,46”
Comment 9: While the limitations are mentioned, they could be discussed with more emphasis on how they specifically influenced the study’s results and what future research should do to overcome these challenges. This includes a more in-depth reflection on the use of tools like ASQ-3, confounders, and potential biases due to the evolving nature of the intervention.
Response 9: Thank you for your feedback. We have now given the direction in which each potential bias might have affected our results, wherever possible (see lines 146-176). We have also recommended in our conclusions, future research that will address some of the highlighted main limitations of our study for producing more robust estimates of the impact of PDC on survival and nutritional and developmental outcomes of at-risk children (see lines 188-192).
Comment 10: provide actionable steps for improving PDC programs, such as integrating more culturally adapted developmental assessment tools or ensuring adequate mentorship across all regions.
Response 10: Thank you for your feedback. We have limited our conclusions/recommendations to the aim of this study – i.e. assessing the impact of PDC intervention on survival, nutritional, and developmental outcomes among at-risk children in rural Rwanda. However, as we discussed our key findings, we also gave examples on how PDC has improved over time. For instance, since 2019, the PDC program replaced the ASQ-3 with the International Guide for Monitoring Child Development – a tool which provides monitoring and early intervention for developmental difficulties and is ideal for use in a primary health care setting as well as been validated in diverse settings (see lines 82-85 in the discussion section).
Reviewer 3 Report
Comments and Suggestions for Authors
Dear Authors:
The study evaluates in detail the stages of the intervention from pre-implementation to post-implementation of PDC in the Kayonza and Kirehe districts.
Introduction:
The introduction presents relevant information for the content of the article.
Materials and Methods: This section is informative and detailed, effectively describing the PDC intervention and its evaluation methodology.
Please mention the criteria you considered when choosing the age of children 11-36 months in Rwinkwavu and 11-47 months in Kirehe at the time of data collection. (lines 139-141)
You also mentioned in lines 144-177 that in Kirehe, children were eligible to be included in the intervention group if they were born between May 2016 and November 2017, which meant they would be between 17 and 39 months old (age adjusted for days of prematurity) at the time of the household survey.
Please clarify this aspect.
On lines 178-179 you mentioned: "In both groups, children with any other comorbidities besides prematurity and low birth weight for Kayonza, and prematurity, low birth weight, and hypoxic-ischemic encephalopathy for Kirehe, were excluded from this study." Please explain why children with HIE were not included in the Kayonza study.
It would have been useful to monitor term low birth weight infants and preterm infants separately - ELBW (birth weight <1000 grams), VLBW (birth weight <1500 grams and >1000 grams), and possibly also LBW (birth weight <2500 grams and >1500 grams). I mention this aspect because survival, nutritional status, and outcomes of neurological and psychosomatic development can vary between degrees of prematurity and also between premature infants and term LBW newborns.
I would like to ask you to mention why you did not consider low birth weight (LBW) newborns with a weight of <2500 grams, not just those with a weight of <2000 grams.
Results and discussions:
The discussion section carefully analyzes the results and links them to the existing literature, reinforcing the significance of the conclusions.
The clear identification of limitations and the provision of context for each limitation, explaining how they may affect the study's conclusions, adds transparency to your research.
Conclusions: represent an effective summary of the main findings of the study, highlighting the positive impact of outpatient intervention on child survival and development.
Moreover, by mentioning the differences between the two districts, the necessity of personalized approaches is emphasized.
It is a well-written article, however, minor revisions are needed before acceptance.
Author Response
Dear reviewer,
Thank you for taking the time to review our manuscript.
Thank you for the opportunity to revise our manuscript. We have responded to each of the comments below.
We feel the helpful feedback from yourself and other reviewers has improved the manuscript.
Sincerely,
Catherine Kirk on behalf of the authors
Response to Reviewer Comments
Comment 1: The study evaluates in detail the stages of the intervention from pre-implementation to post-implementation of PDC in the Kayonza and Kirehe districts.
Response 1: Thank you for your feedback.
Introduction:
Comment 2: The introduction presents relevant information for the content of the article.
Response 2: Thank you.
Materials and Methods:
Comment 3: This section is informative and detailed, effectively describing the PDC intervention and its evaluation methodology.
Response 3: Thank you.
Comment 4: Please mention the criteria you considered when choosing the age of children 11-36 months in Rwinkwavu and 11-47 months in Kirehe at the time of data collection.
Response 4: Thank you for pointing this out. We have now clarified in the methods that this was the target age group for household surveys, where children were included in surveys if they would be aged between 11-36 months and 11-47 months at the time of data collection in Rwinkwavu and Kirehe, respectively (see lines 145-146). The reason for extended age in Kirehe was also given and reads as follows “The extended age range in Kirehe was included due to anticipated lower participation in the household survey, particularly among pre-intervention group, as it was previously observed in the Rwinkwavu sample (see lines 146-150)”. We have also added additional information that the target age group for survey participants was determined based on the timeline of household surveys for both control and intervention groups in each district, as well as the need to allow at least one year of exposure to the intervention for the PDC group (see lines 149-152).
Comment 5: You also mentioned in lines 144-177 that in Kirehe, children were eligible to be included in the intervention group if they were born between May 2016 and November 2017, which meant they would be between 17 and 39 months old (age adjusted for days of prematurity) at the time of the household survey. Please clarify this aspect.
Response 5: Thank you for this comment. We have now clearly presented that the date of birth and the same age intervals at the time of household surveys for both control and intervention groups were among the additional eligibility criteria that we retrospectively defined to create more comparable groups for this study (see lines 181-189).
Comment 6: On lines 178-179 you mentioned: "In both groups, children with any other comorbidities besides prematurity and low birth weight for Kayonza, and prematurity, low birth weight, and hypoxic-ischemic encephalopathy for Kirehe, were excluded from this study." Please explain why children with HIE were not included in the Kayonza study.
Response 6: Thank you for this comment. The reason why HIE children were not included in the Kayonza/Rwinkwavu sample was given in the methods (see lines 138-144) and reads as follows: “At the beginning of the PDC implementation in Kayonza, the baseline survey (historical control group) included only preterm or LBW children as this was anticipated to be the primary and largest population served by the new service (PDC). However, by the time of expansion into Kirehe, it was understood that children with birth asphyxia and suspected HIE were a comparably large population served by the clinic, representing about a third of children enrolled in PDC, and therefore were included in data collection to assess programmatic impact.”
Comment 7: It would have been useful to monitor term low birth weight infants and preterm infants separately - ELBW (birth weight <1000 grams), VLBW (birth weight <1500 grams and >1000 grams), and possibly also LBW (birth weight <2500 grams and >1500 grams). I mention this aspect because survival, nutritional status, and outcomes of neurological and psychosomatic development can vary between degrees of prematurity and also between premature infants and term LBW newborns.
Response 7: Thank you. We agree with your comment that the patterns of survival, nutritional and developmental outcomes can vary by diagnosis, gestational age and size at birth. For example, we conducted a sensitivity analysis to separately report on estimated impact of PDC on preterm/LBW children, as we hypothesized that their survival, nutritional and developmental outcomes might be different from those of HIE children (see the supplementary table 1). However, some of the sub-analyses are not possible due to small sample sizes in some sub-categories. We have recommended further research on the impact of PDC that will address some of these limitations in the future.
Comment 8: I would like to ask you to mention why you did not consider low birth weight (LBW) newborns with a weight of <2500 grams, not just those with a weight of <2000 grams.
Response 8: Thank you for pointing this out. We have now clarified in the methods that LBW diagnosis for admission in NCU was defined as a birthweight<2,000 grams (see lines 137-138). Children who were LBW, but greater than 2,000 grams, would only be eligible for the PDC if they had another reason for referral (e.g., preterm, HIE, etc.).
Results and discussions:
Comment 9: The discussion section carefully analyzes the results and links them to the existing literature, reinforcing the significance of the conclusions.
Response 9: Thank you for this feedback.
Comment 10: The clear identification of limitations and the provision of context for each limitation, explaining how they may affect the study's conclusions, adds transparency to your research
Response 10: Thank you so much for this feedback.
Comment 11: Conclusions: represent an effective summary of the main findings of the study, highlighting the positive impact of outpatient intervention on child survival and development.
Response 11: Thank you so much for this feedback.
Comment 12: Moreover, by mentioning the differences between the two districts, the necessity of personalized approaches is emphasized.
Response 12: Thank you so much for your feedback.
Comment 13: It is a well-written article, however, minor revisions are needed before acceptance.
Response 13: Thank you very much for your feedback. Your review comments have helped us to make more improvement on this paper.
Round 2
Reviewer 2 Report
Comments and Suggestions for Authors
Well, they were revised.